# *Polianthes tuberosa*-Mediated Silver Nanoparticles from Flower Extract and Assessment of Their Antibacterial and Anticancer Potential: An In Vitro Approach

**DOI:** 10.3390/plants12061261

**Published:** 2023-03-10

**Authors:** Mousa A. Alghuthaymi, Sunita Patil, Chandrasekaran Rajkuberan, Muthukumar Krishnan, Ushani Krishnan, Kamel A. Abd-Elsalam

**Affiliations:** 1Biology Department, Science and Humanities College, Shaqra University, Alquwayiyah 11971, Saudi Arabia; 2Rajiv Memorial Education Society’s College of Pharmacy, Gulbarga 585102, India; 3Department of Biotechnology, Karpagam Academy of Higher Education, Coimbatore 641021, India; 4Department of Petrochemical Technology, Anna University, Tiruchirappalli 620024, India; 5Karpaga vinayaga College of Engineering, Chengalpattu 603308, India; 6Plant Pathology Research Institute, Agricultural Research Center, Giza 12619, Egypt

**Keywords:** silver nanoparticles, synthesis and characterization, anticancer activity, flower, *P. tuberosa*

## Abstract

Plant-mediated metallic nanoparticles have beenreported for a diversified range of applications in biological sciences. In the present study, we propose the *Polianthes tuberosa* flower as a reducing and stabilizing agent for the synthesis of silver nanoparticles (PTAgNPs). The PTAgNPs were exclusively characterized using UV–Visible spectroscopy, Fourier transform infrared spectroscopy (FTIR), scanning electron microscopy (SEM), X-ray diffraction (XRD), atomic force microscopy, zeta potential, and transmission electron microscopy (TEM) studies. In a biological assay, we investigated the antibacterial and anticancer activity of silver nanoparticles in the A431 cell line. The PTAgNPs demonstrated a dose-dependent activity in *E. coli* and *S. aureus*, suggesting the bactericidal nature of AgNPs. The PTAgNPs exhibited dose-dependent toxicity in the A431 cell line, with an IC_50_ of 54.56 µg/mL arresting cell growth at the S phase, as revealed by flow cytometry analysis. The COMET assay revealed 39.9% and 18.15 severities of DNA damage and tail length in the treated cell line, respectively. Fluorescence staining studies indicate that PTAgNPs cause reactive oxygen species (ROS) and trigger apoptosis. This research demonstrates that synthesized silver nanoparticles have a significant effect on inhibiting the growth of melanoma cells and other forms of skin cancer. The results show that these particles can cause apoptosis or cell death in malignant tumor cells. This suggests that they could be used to treat skin cancers without harming normal tissues.

## 1. Introduction

Since the mid-20th century, climate change has badly hampered human health, and at the same time, incidences of skin cancer have increased at an alarming rate due to ozone layer depletion, UV radiation exposure, and an increase in global temperature [1,2]. In terms of incidence, prevalence, and disability-adjusted life years, the global burden of skin cancer is increasing in a way that is not the same for all age groups [3]. At the current epoch, there are two types of skin cancer: melanoma and non-melanoma. The incidence of non-melanoma skin cancers in Caucasians is 18–20 times higher than that of melanoma skin cancers [4]. Non-melanoma skin cancer comprises squamous cell carcinoma (SCC) and basal cell carcinoma (BCC). Globally, two to three million cases of non-melanoma skin cancer occur, and SCC accounts for 20% of all non-melanoma skin cancers; SCC can metastasize and cause devastating outcomes [5]. The non-melanoma skin cancer disease burden will continue to increase or it will be stable at a higher level in the coming years. As a result, appropriate policies should be put in place to manage skin cancer [6]. Surgery or radiotherapy can cure BCCs and SCCs, but certain cancers and patient-related factors such as patient co-morbidities, high tumor multiplicity, metastasis, and patient preference make these therapies inadequate or inappropriate. Systemic therapy is, however, severely constrained by aged patient populations and the treatment toxicity of systemic therapy in specific high-risk patient populations [7]. Due to the emergence of chemotherapeutic drug resistance, the discovery of effective new medications is of utmost relevance in the treatment of cancer [8]. A powerfully effective treatment with high antitumor efficiency and decreased systemic adverse effects was also required because certain anticancer medicines were linked to side effects and high toxicity [9]. It is time to reevaluate the use of natural products in cancer treatment given modern technology and cancer therapies. Currently, the anticancer potential of several natural compounds is being studied.

Natural products have made significant contributions to cancer treatment; today, scientific advances offer up new avenues for cancer treatment with innovative biological constituent-based compounds [10]. Biological nanoparticles have recently been widely explored for their antibacterial, antioxidant, and anticancer properties [11]. Nanoparticles may easily interact with cellular biomolecules, resulting in improved signals and target specificity for cancer detection and therapy. Conventionally, nanoparticles can be fabricated through chemical and physical methods. Both methods have multifaceted distinct properties, but they are toxic in biomedical applications [12]. The synthesis of nanoparticles is gaining momentum dueto their stupendous properties and lack of toxicity. Green nanoparticles synthesized from plants, fungi, bacteria, and algae exhibit cytotoxicity against diverse types of cancer [7,13]. The biological synthesis of silver nanoparticles is an easy, economic, and eco-friendly process in which plants extract constituents to efficiently reduce silver ions to silver atoms to produce silver nanoparticles (1–100 nm). Plant constituents can increase the stability of silver nanoparticles, which have high efficacy and low toxicity [14,15]. AgNPs have been successfully evaluated in various cancer diseases and can be developed as a drug in the near future. AgNPs cause cytotoxicity through a variety of mechanisms including silver ion formation and radical formation, which cause the deregulation of various cellular operations, resulting in cellular damage and death [16].

The Agavaceae family of flowering plants grows in hot, dry climates. Tuberose, scientifically known as *Polianthes tuberosa*, is an Asparagaceae family, and the plant is bestowed with medicinal properties. Generically, the *P. tuberosa* flower contains aromatic essential oils that are used intensively in perfume and cosmetics. Aside from the aforementioned feature, the flowers’ major metabolites are terpenoid derivatives [17]. Geraniol has been shown to have anticancer activity against various types of cancer by modulating various signaling pathways [18]. Silver nanoparticles possess excellent antibacterial activity; likewise, AgNPs reportedly had an effective action against cancer cells, the targeted localization of drugs in the cancer cells, and diagnosing the cancer cells. Previously, we have reported the anticancer efficacy of *P. tuberosa* gold nanoparticles against the cancer cell line MCF 7 [19]. The present study was designed to synthesize silver nanoparticles from an aqueous extract of the flower *P. tuberosa*. Additionally, to evaluate the bactericidal activity, PTAgNPswereexamined against *E. coli* and *S. aureus*. Furthermore, the aforementioned silver nanoparticles were studied for their anticancer attributes in a skin cancer cell line. The findings of the study will have a significant impact on the research fraternity’sefforts to discover more intriguing nanoparticles to mitigate skin cancer.

## 2. Results

To synthesize PTAgNPs, we followed a decisive method by employing a flower extract and challenging it with a silver nitrate solution in a simple step. Upon reaction initiation, the synthesis process wasaccelerated, and finally, a dark brownish color wasobserved (Figure 1).

The color change wasdue to the reaction of functional chemical moieties in *P. tuberosa* with silver nitrate. We optimized the following parameters for the fine generation of PTAgNPs: flower extract (10–100 mL); silver nitrate (1–5 mM); temperature (40–90 °C); pH (4, 7, 9); and time (15 min–5 h). After optimizing the various parameters, we fixed the reaction parameters as follows: flower extract 30 mL; silver nitrate 1 mM; 90 min; pH 9; temperature 60 °C; stoichiometric ratio (30 mL flower extract + 70 mL AgNO_3_). In our study, we observed that a pH of 9 favors the synthesis of silver nanoparticles. The ionization behavior of tannins and phenolic chemicals in the flower extract of *P. tuberosa* causes the nucleation process in the formation of AgNPs. When silver nanoparticles were created using the husk of an oak fruit, similar pH measurements were made [20]. In our investigation, the total production time for nanoparticles was close to five hours. The size of the nanoparticles causes an increase in reaction time. The amount of silver nitrate present, plant extracts, temperature, and pH all affect how quickly silver ions are synthesized. Similarly, the synthesis of silver nanoparticles from the seed extract *Pimpinella anisum* took 96 h for complete synthesis and stabilization [21]. Initially, the PTAgNPs were analyzed in UV–Visible spectroscopy to ascertain the size and shape of the nanoparticles in an aqueous suspension. UV–Visible spectroscopy measures absorbance at 415 nm, which is due to the phenomenon of surface plasmon resonance (SPR) [22]. Due to the optical resonant property SPR, the PTAgNPs strongly absorbed electromagnetic waves in the visible light region (370–450 nm) in the current study [23]. In Figure 2, a single SPR band was observed from the UV–Vis analysis, where it can be inferred that the synthesized PTAgNPs might have a spherical shape.

FTIR investigation revealed the interaction of plant chemical moieties as a reducing and stabilizing agent in the formation of PTAgNPs (Figure 3). The FTIR spectrum of PTAgNPs expressed visible bands at 665 cm^−1^, 1031 cm^−1^, 1244 cm^−1^, 1516 cm^−1^, 1550 cm^−1^, 1643 cm^−1^, 1735 cm^−1^, 2325 cm^−1^, and 2921 cm^−1^. The band 665 cm^−1^ can be assigned to the –C–H bend (alkane); 1031 cm^−1^ and 1244 cm^−1^ were assigned to the–C–N stretch (aliphatic amines); 1516 cm^−1^ and 1550 cm^−1^ correspond to the =C–H aromatics; the peak at 1643 cm^−1^ directs to the –N–H bend (amide), the peaks observed at 1735 cm^−1^, 2325 cm^−1^, and 2921 cm^−1^ are postulated to be–C=O stretch (carboxyl) and C=C stretching alkyne [24]. From the spectrum, it is inferred that proteins, alkaloids, flavonoids, and tannins might be the responsible chemical groups thatact as reducing and stabilizing agents in the formation of silver ions [25].

The zeta potential of PTAgNPs was determined to be −21 mV and stable (Figure 4). In general, nanoparticles in solution will interact with one another through van der Waals forces and have the propensity to aggregate quickly [26]. This effect is influenced by the nanoparticles’ surroundings and surface charge (zeta potential). Low zeta potential particles will group, but high zeta potential particles will remain stable over an extended time [27].

The crystallinity of PTAgNPs was ascertained by XRD analysis. In Figure 5, the 2-theta peaks observed at 38.18°, 44.47°, 64.48°, 77.48°, and 81.78° corresponded to the Bragg reflections in the 111, 200, 220, 311, and 222 planes, respectively [28]. From the JCPDS (04-0784) analysis, the synthesized PTAgNPs were in the FCC structure. Apart from these, a few other sharp peaks were also seen, which might be due to the existence of the organic phytochemicals in the flower extract [29].

The morphology and its associated features were investigated by scanning electron microscopy (SEM), transmission electron microscopy (TEM), and atomic force microscopy (AFM). In Figure 6, the SEM images of PTAgNPs showed spherical-shaped nanoparticles with varying sizes ranging from 13 to 87 nm. The SEM image revealed the interaction of bioorganic molecules with silver nanoparticles by hydrogen bonding or electrostatic interaction [30]. The transmission electron micrograph of PTAgNPs shows spherical-shaped particles with a size distribution ranging from 40 to 70 nm (Figure 6). The elemental composition of PTAgNPs revealed the presence of silver as a major constitutional element (Figure 6) in the EDX spectrum. The 2D and 3D AFM images (Figure 7a–c) infer that the synthesized nanoparticles were in the size regime of 21–80 nm with spherical shapes. Thus, from the exclusive microscopic studies, it can beinferred that synthesized PTAgNPs were spherical and polydispersed.

The antibacterial activity of PTAgNPs was determined by the agar well diffusion method. The results of the study were presented as a graph (Figure 8). As expected, the PTAgNPs exerted superior activity against the tested pathogens. In contrast, we observed that PTAgNPs exerteda stronger activity than *P. tuberosa* gold nanoparticles, as previouslyreported [19]. This is due to the bactericidal nature of silver and the phytoconstituents that coat the surface of the nanoparticles [31]. The bactericidal activity of the nanoparticles will differ with the microbes, metal/metal oxide nanoparticles and phytoconstituents. However, in the present study, the PTAgNPs showcased the potential attributes of inhibiting the growth of pathogens *Escherichia coli* and *Staphylococcus aureus subsp. aureus* in a dose-dependent manner. The PTAgNPs at various concentrations sharply inhibited the pathogens when compared with the control gentamicin. The antibacterial nature of silver nanoparticles is directly influenced by the composition of the cell wall [32]; in the present study, we observed that *E. coli* wasmore highly inhibited than *S. aureus*. This is due to the key component of the peptidoglycan layer; the Gram-negative *E. coli* peptidoglycan layer is about 3–4 nm thickness where it is prone to the attack of chemicals [33]. In addition, the Gram-negative *E. coli* cell wall contains lipopolysaccharides thatpromote theadhesion of silver nanoparticles [34]. In the investigated Gram-positive bacteria, the peptidoglycan layer (30–40 nm) thickness resisted the interaction of AgNPs. However, the antibacterial activity of AgNPs is well-established in various pathogens as reported elsewhere. Nevertheless, some hypotheses have been put forward regarding the mechanistic aspects of silver nanoparticles. The cascade of mechanistic events can becategorized into four phases. Phase I involves the interaction and adhesion of silver nanoparticles with the peptidoglycan layer of the cell membrane. It alters the membrane structure and permeability and impairs the activity of transport. In phase II, the silver nanoparticles penetrate inside the cell membrane and damage the intracellular organelles, cause mitochondrial dysfunction, denature the proteins, destabilize the ribosomes, and intercalate the DNA. In phase III, the silver nanoparticles induced ROS and caused the oxidization of proteins, lipids, and DNA bases, while inphase IV, the generation of ROS induces the cell signaling pathways to activate the apoptosis process [35]. To confirm the above hypothesis, Gopinath et al. [36] synthesized biogenic nanoparticles from *Pseudomonas putida MVP2*; the synthesized nanoparticles were evaluated for antibacterial activity against *Helicobacter pylori*, *Escherichia coli*, *Bacillus cereus*, *Staphylococcus aureus*, and *Pseudomonas aeruginosa*. The SEM analysis after 60 min of treatment caused the cell membrane damage, and integrity followed by the increased production of ROS demonstrated by the LDH assay. The above study confirmed the testament that AgNPs cause cell membrane damage > intracellular leakage > ROS > oxidative stress > apoptosis > cell death. Thus, the above mode of action might be the reason for the antibacterial activity of AgNPs.

The cytotoxic effect of PTAgNPs was evaluated in cell line A431 by using the MTT assay in a dose-dependent concentration method. In Figure 9, the effect of PTAgNPs in cell line A431 with the time response was plotted. PTAgNPs clearly exhibited a dose-dependent activity, causing 100% mortality in cell line A431 at 24 and 48 h, respectively, with an IC_50_ of 54.56 µg/mL. The cellular effect of PTAgNPs in the A431 cell line is primarily due to the surface coating agent, size, surface charge, solubility, and uptake of AgNPs into the cells and cell line [37]. Generally, the AgNPs trigger anticancer activity that will vary from different cell lines and other factors. The cytotoxicity of AgNPs must be clearly understood before its useas a drug in the future. Various scientific evidence hasbeen reported regarding the cytotoxicity of silver nanoparticles against various cancer cell lines. Initially, the AgNPs are uptakeninto the cells by the process of pinocytosis, endocytosis, and phagocytosis [38]. Once AgNPs enter the cells, it reacts with certain biomolecules (DNA/RNA/proteins/lipids) and induces oxidative stress, which subsequently leads to the production of ROS [39]. Oxidative stress is caused by mitochondrial dysfunction. The increased production of ROS damages the mitochondrial respiratory chain and eventually leads to DNA damage [40]. The ROS induces DNA damage at apurinic/apyrimidinic (AP) sites, single-strand breaks, DNA bases, base lesions, chromosomal aberrations, and mutations [41]. As a result of DNA damage, the inflammatory cytokines IL-1, IL-6, and TNF-α IL-1β are expressed, which indicates inflammasome activation [38]. After the severity of DNA damage and the increased expression of inflammatory biomarkers, cytochrome c was released from the cell and Bax was translocated to the mitochondria, which induces the apoptosis process [42]. Therefore, in the present study, we speculate that PTAgNPs might induce the apoptosis process by the above mechanistic process. Biogenic-mediated metallic nanoparticles in recent reports have documented the cytotoxic properties in normal cell lines. The report byKarimzadeh et al. [43] proved that biogenic silver nanoparticles synthesized from *Oxalis corniculata* did notexert much toxicity on the L929 normal fibroblast cells. Likewise, biogenic gold nanoparticles synthesized from *Sesuviumportulacastrum* L. did notimpose much toxicity on the HBL100 cells [44]. From the above reports, we speculate that the flower-mediated silver nanoparticles will not impose toxicity on normal cells and will be compatible. Nevertheless, the molecular mechanistic nature of the biogenic nanoparticles needs to be studiedin various cell lines and in vivo models.

In addition, flow cytometry was performed to evaluate the molecular process in response to PTAgNP therapy in cell line A431. The cell cycle analysis of control (a) and treated (b) cells is shown in Figure 10. After 24 h, the control cells had 30.95 percent of their cells in the S phase, whereas the treated cells had 50.09 percent of their cells in the S phase. Similarly, the proportion of control cells in the G2/M phase was 4.30 percent, while it was 14.63 percent in the treated cells. These findings demonstrate that cells are severely halted during the S phase, andinhibiting the elongation of DNA replication during the S phase instructs the replication checkpoint to trigger cell death [45,46].

The Comet assay determines the DNA damage caused by PTAgNPs in the A431 cells. In Figure 11, PTAgNPs caused DNA damage severity and tail length in the A431 cell line. The percentage of tail length in the treated cell line was 39.9% ± 18.15, and no diffusion of DNA fragments was observed in the untreated cells. The phenomenon of tail damage could be attributed to the induction of intracellular ROS production, which caused oxidative stress in the A431 cells, causing nuclear condensation and chromosomal DNA fragmentation, ultimately leading to apoptosis [47]. Overall, the assay demonstrated that PTAgNPs initiate DNA fragmentation caused by apoptosis, which suggests that PTAgNPs cause genotoxicity and mutagenicity in the DNA of cancer cells.

Following the evaluation of cytotoxicity and genotoxicity assays, we investigated the intracellular production of ROS upon the treatment of PTAgNPs in the A431 cell line (Figure 12a–c). For analysis, the cells were treated with the IC_50_ of PTAgNPs and stained with dichlorofluorescin diacetate (DCFH-DA). In response to the PTAgNP treatment in the A431 cell line, the control cells emitted a low fluorescence signal while the PTAgNP treated cells exhibited a higher green fluorescent intensity, which indicates the overproduction of ROS. PTAgNPs trigger ROS, followed by oxidative stress; subsequently, the cells will respond by activating pro-inflammatory signaling cascades and inducing cell death by apoptosis [48].

## 3. Materials and Methods

### 3.1. Chemicals and Reagents

Merck Limited supplied the (99.0%) silver nitrate. The A-431 melanoma cell line was obtained from NCCS Pune and kept in HIMEDIA in Dulbecco’s modified Eagle medium (DMEM). Deionized water was used for the preparation of chemicals and reagents.

### 3.2. Synthesis of Flower-Mediated Silver Nanoparticles

The green synthesis approach was used to create *P. tuberosa* flower-mediated silver nanoparticles. Flowers were acquired fresh and uninfected at a local market in Coimbatore, India. The flowers were brought to the lab soon after being collected and washed with water. The flowers were dried in a hot air oven at 37 °C for 7 days after washing with sterilized water. Following the incubation period, the dried flowers were ground into a fine powder. The powder was employed in the synthesis process.

For the synthesis, 50 g of flower powder was poured in 250 mL of distilled water and stirred for 2 h in a magnetic stirrer at 60 °C. For fine synthesis, we optimized the reaction kinetics of the flower extract (10–100 mL), silver nitrate (1–5 mM), temperature (40–90 °C), pH (4, 7, 9), and time (15 min–5 h). Based on the observations, we fixed the final reaction kinetics as follows: flower extract 30 mL; silver nitrate 1 mM; 90 min; pH 9; temperature 60 °C; plant: salt ratio (30 mL flower extract + 70 mL AgNO_3_). The above reaction kinetics proceeded for the fine synthesis of PTAgNPs. After the incubation time, the synthesized nanoparticles were visually observed for color change and subjected to further characterization. After synthesizing, the nanoparticles were centrifuged and washed with methanol to remove the impurities.

### 3.3. Characterization of Flower-Mediated Silver Nanoparticles

The synthesized PTAgNPswere characterized to determineits size, morphological character, and crystallinity [49]. Synthesis of PTAgNPs was confirmed by UV–Visible spectroscopy (Shimadzu UV-2450) carried out at a 1 nm resolution in the 200–800 nm wavelength range at 25 °C. Fourier transmission infrared spectroscopy (FTIR) using an affinity^−1^ Shimadzu FTIR was performed using potassium bromide in 1:200 proportions with the plant extract and silver nanoparticles separately in the spectral range of 4000–500 cm^−1^, X-ray diffraction(XRD) using an X’Pert Pro X-ray diffractometer using 40 kV, 40 mA X-ray source with Cu Kα1 radians in diffraction angle 0–90°, field emission scanning electron microscope (FESEM), and energy dispersive X-ray spectra (EDX) using FEI—QUANTA–FEG 250 at 30 kV. FESEM micrographs were captured at different magnifications by focusing on different fields. HRTEM analysis was performed using the JEOL JEM 2100 instrument, JEOL Instruments USA For microscopy analysis, the sample was prepared by coating PTAgNPs on 300 mesh size copper grids at different magnifications at an accelerating voltage of 300 kV, and the zeta potential was determined using a Malvern particle size analyzer at the angle of 90° and temperature of 25 °C; before the analysis, the synthesized SNPs were diluted (5-fold) and sonicated at 50 W for 3 min repeatedly.

### 3.4. Antimicrobial Activity of PTAgNPs

The antibacterial activity of PTAgNPs was assessed by the agar diffusion method. For the assay, the bacterial pathogens *Escherichia coli* (MTCC no.: 443) and Gram-positive *Staphylococcus aureus* subsp. *aureus* (MTCC no.: 737) were procured from the Institute of Microbial Technology, Chandigarh, India. The cultures were initially inoculated in nutrient agar medium and kept forovernight incubation at 37 °C. Later, the cultures were inoculated in the freshly prepared Mueller–Hinton agar (MHA); using a borer, a well with a diameter of 6 mm was punctured in the agar medium. With a sterile cotton swab, the test cultures were uniformly swabbed in the agar surface uniformly and smoothly. Then, the PTAgNPs with different concentrations (20, 40, 60 µg/mL) were added into the well and kept for incubation at 37 °C for 24 h. After the incubation time, the plates were observed for the zone of inhibition (ZOI) and calibrated using a vernier caliper in mm.

### 3.5. In Vitro Anticancer Activity of Silver Nanoparticles

#### 3.5.1. MTT Assay

The A431 cells (10,000/well) were plated in two 96-well plates. The cells reached confluence after 24 h of incubation. The grown culture was treated with different concentrations of PTAgNPs (10 to 100 μg/mL). The control and blank were maintained throughout the experiment. The plate was incubated in a 5% CO_2_ incubator in a humidified atmosphere at 37 °C for 24 h and 48 h. The cells were washed with PBS and then treated with 100 µL of MTT (5 mg/mL) solution and incubated at 37 °C for 4 h. The MTT dye was removed and DMSO 100 µL was added to the culture. Then, the optical density was read at 576 nm. The effect of the PTAgNPs on the A431 cells and % viability was determined by comparing the results with the control.

#### 3.5.2. Cell Cycle Analysis

The cell cycle analysis was performed to determine the cell cycle arrest phase after PTAgNP treatment [50]. The A431 cells were seeded in six-well plates with MEM supplemented with 10% FBS. The cells were incubated for 24 h under standard cell culture conditions. Then, three wells of cell culture weretreated with a 56 µg/mL concentration of PTAgNPs, and three untreated wells served as the controls. After 24 h, the cells were trypsinized and washed twice with PBS, and then the cells were suspended again in stain PI (200 μg/mL DNase free RNase, 4 mM sodium citrate, 50 μg/mL propidium iodide, 0.1% Triton X-100) and incubated for 15 min. The experiment was performed in flow cytometry.

#### 3.5.3. Comet Assay

After treatment with PTAgNPs, the A431 cells were re-suspended in ice-cold PBS [51]. In a glass slide, approximately, 10,000 cells in a volume of 100 mL were coated with a thin layer ona 1.0% (*w*/*v*) agarose frosted glass slide and covered with a coverslip and incubated in cold condition for 10 min. After the incubation period, the coverslips were removed and the slides were immersed in an ice-cold lysis solution containing 2.5 M NaCl, 10 mM Tris, 1% (*w*/*v*) *N*-lauroyl-sarcosine, 100 mM Na2-EDTA, and 1.0% Triton X-100. Electrophoresis was performed for 30 min at 300 mA with the slides inserted into a horizontal electrophoresis tank filled with buffer (0.3 M NaOH, 1 mM EDTA, pH 13). After the process, slides were washed with a 0.4 M Tris-HCl solution, and ethidium bromide was added. After staining for 5 min, the slides were subjected to microscopic examination, and the images were captured and recorded.

#### 3.5.4. Determination of ROS Level

The cells were seeded into 96-well plates at a density of 5000–10,000 cells per well. After 24 h, cells were washed with PBS buffer and treated with 100 μM DCFH-DA for 1 h [52]. Cells were again washed with PBS and treated with the IC_50_ concentration of PTAgNPs and untreated wells were considered as the control. After incubation for 6 h, the cells were rinsed with PBS and 2 mL of PBS was added to each well, and the DCF fluorescence intensity was examined with a spectrofluorometer emission at 530 nm and excitation at 485 nm. The fluorescence was imaged with a fluorescent microscope (Olympus CKX41 with Optikapro5 CCD camera, Olympus Microscopes, Westborough, MA, USA).

#### 3.5.5. Statistical Analysis

All of the studied experiments were performed in triplicate and expressed as ±standard deviation. The MTT assay yielded a percentage of viability, with concentrations expressed to the control viability. The obtained results were statistically analyzed using the Student’s *t*-test, with *p* ≤ 0.05 considered significant.

## 4. Conclusions

The therapeutic effectiveness of silver nanoparticles in the skin cancer cell lines is described in the current work. The Ag nanoparticles were produced using a straightforward one-pot process, and TEM, SEM, and AFM were used for their analysis. The characterization investigations showed that crystalline, spherical, and nanoscale particles were present. The bactericidal activity of PTAgNPs was examined against *E. coli* and *S. aureus*. The PTAgNPs caused significant lethal effects against the pathogens in a dose-dependent manner. Furthermore, PTAgNPs caused toxicity in the A431 cell line in a dose-dependent manner with an IC_50_ of 56.54 µg/mL; flow cytometry analysis showed that the S phase was arrested. The ROS analysis further confirmed that PTAgNPs caused cell death in the A431 cell line by inducing ROS. Therefore, after being submitted to in vivo research, it can be concluded, based on the experimental data, that silver nanoparticles are a potential agent for chemotherapy. In summary, the current study demonstrated promising results regarding the use of silver nanoparticles against skin cancers through its ability to arrest S phase progression while causing toxicity on the A431 cells by the induction of oxidative stress response pathways, ultimately leading toward apoptosis, thus making it a potentially viable option for future therapies targeting this particular form of disease, if proven to be successful after going through the necessary steps before being approved for human usage.

## Figures and Tables

**Figure 1 plants-12-01261-f001:**
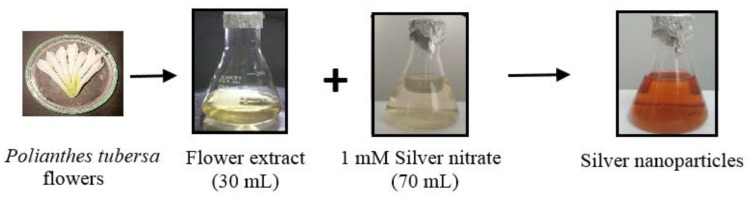
Synthesis of silver nanoparticles from the *P. tuberosa* flower extract.

**Figure 2 plants-12-01261-f002:**
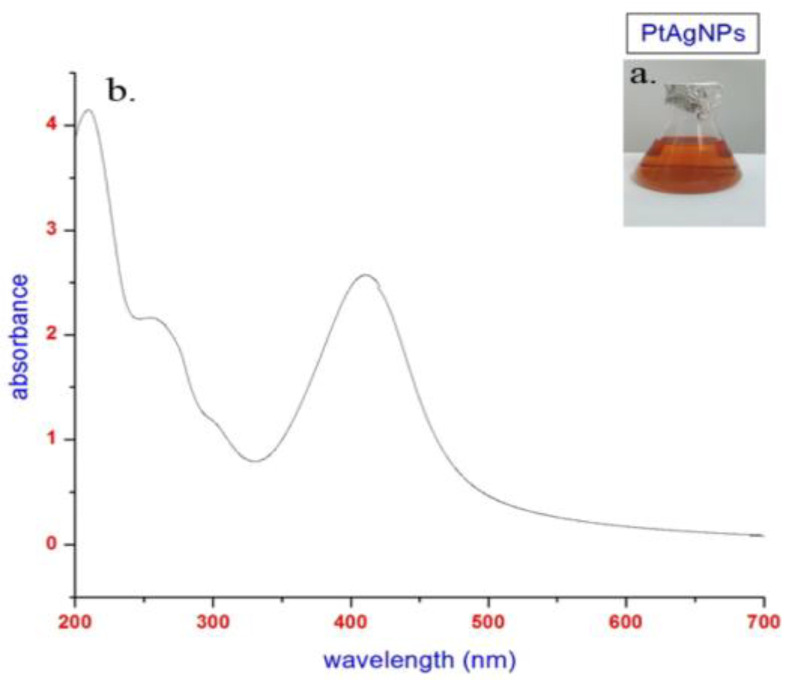
(**a**) Color change of PTAgNPs. (**b**) UV–Vis spectrum of PTAgNPs measuring absorbance at 415 nm.

**Figure 3 plants-12-01261-f003:**
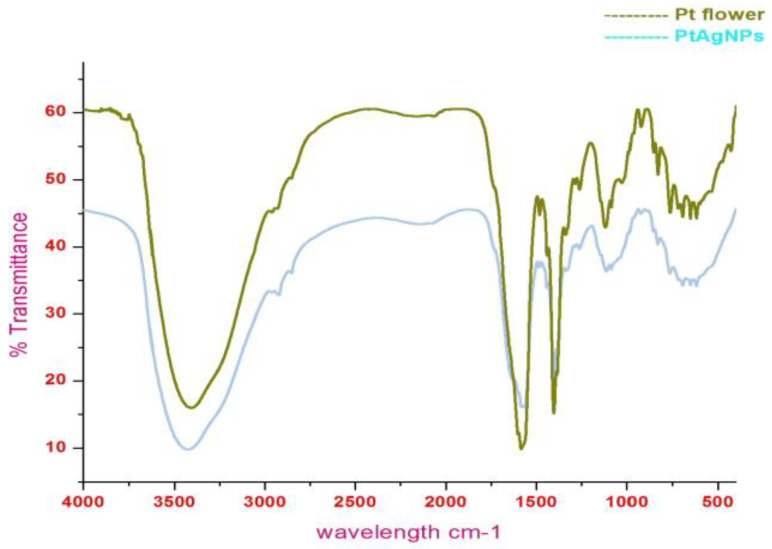
FTIR spectrum of the PTAgNPs and *P. tuberosa* flower extract (powder).

**Figure 4 plants-12-01261-f004:**
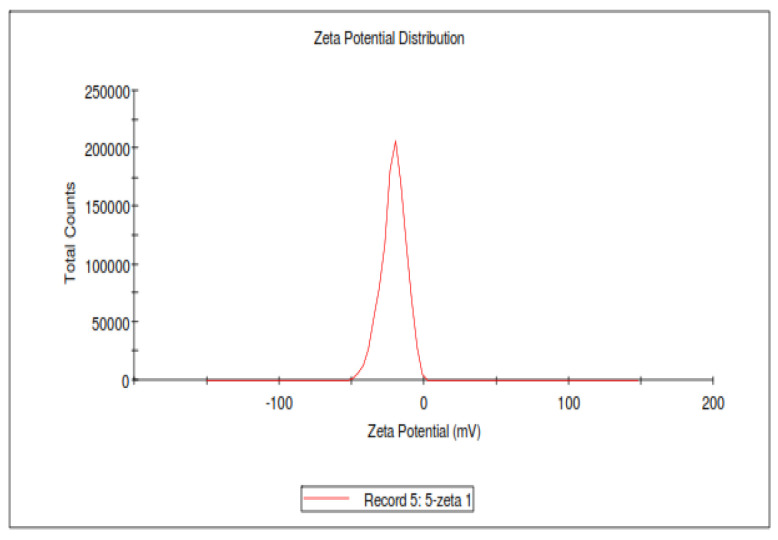
Zeta potential of PTAgNPs.

**Figure 5 plants-12-01261-f005:**
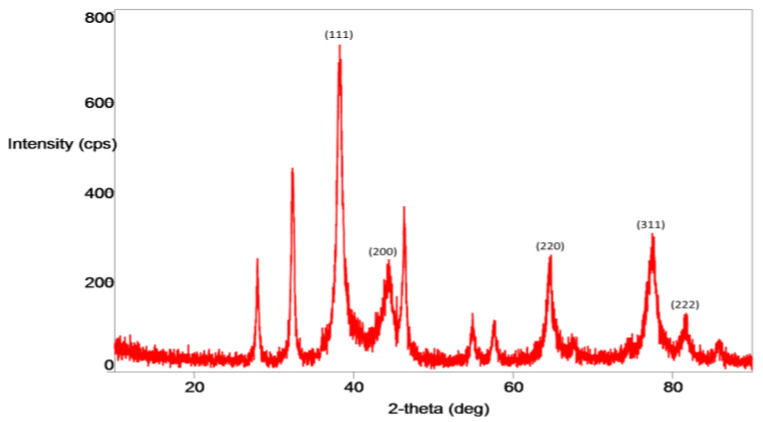
XRD spectrum of the PTAgNPs.

**Figure 6 plants-12-01261-f006:**
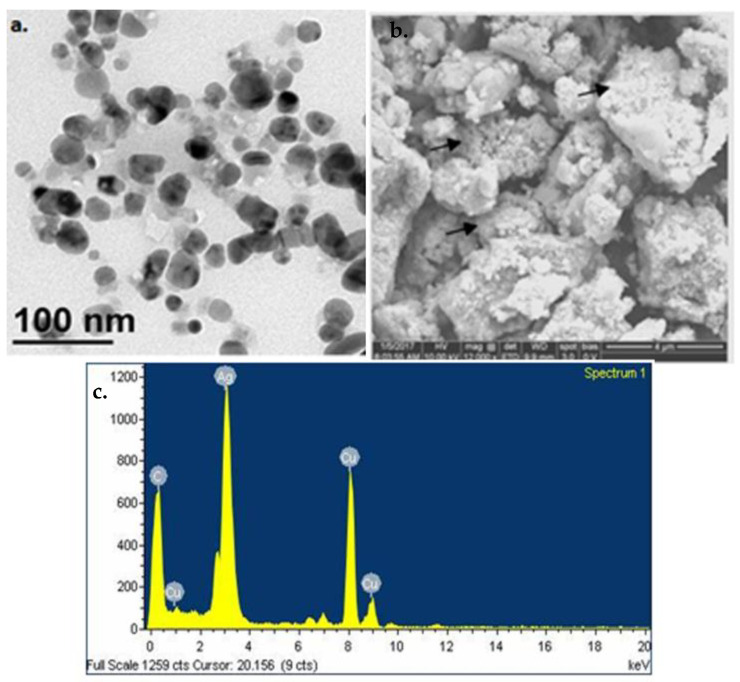
(**a**) TEM microscopic images of thePTAgNPs. (**b**) SEM images of the PTAgNPs. (**c**) EDX spectrum of the PTAgNPs.

**Figure 7 plants-12-01261-f007:**
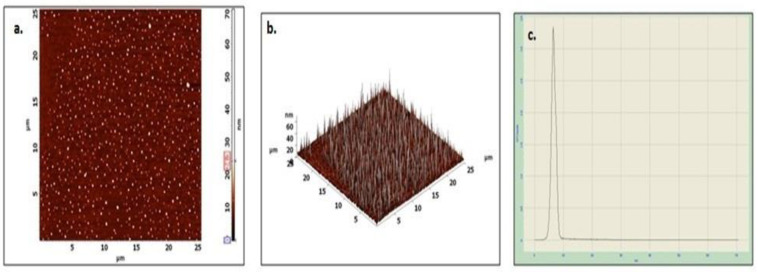
(**a**–**c**) AFM images of the PTAgNPs.

**Figure 8 plants-12-01261-f008:**
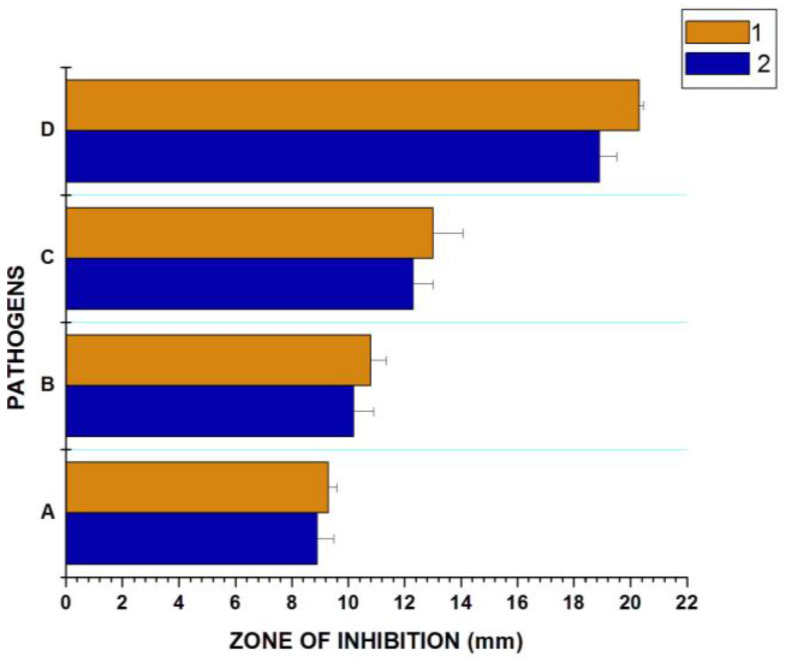
Antibacterial activity of PTAgNPs. 1—*E. coli*, 2—*S. aureus*. (**A**) 25 µg/mL; (**B**) 50 µg/mL; (**C**) 75 µg/mL; (**D**) control gentamicin 20 µg/mL.

**Figure 9 plants-12-01261-f009:**
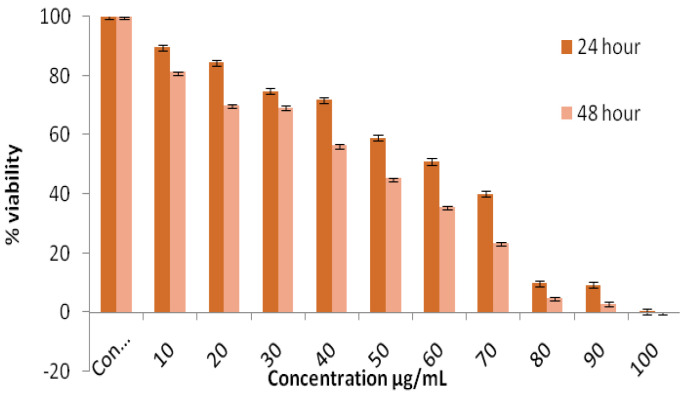
Cell viability assay of the PTAgNPs validated by the MTT method.

**Figure 10 plants-12-01261-f010:**
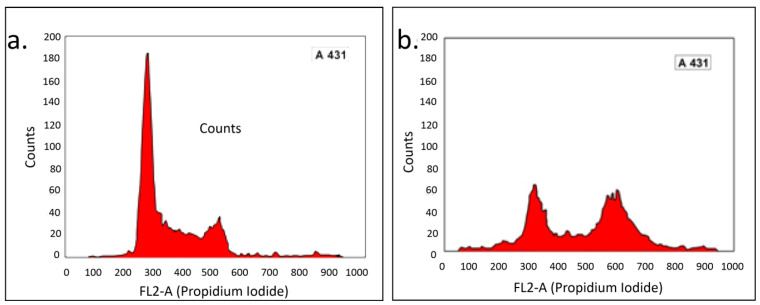
Cell cycle analysis of PTAgNPs in response to the A431 cells by flow cytometry. (**a**) Control; (**b**) treated.

**Figure 11 plants-12-01261-f011:**
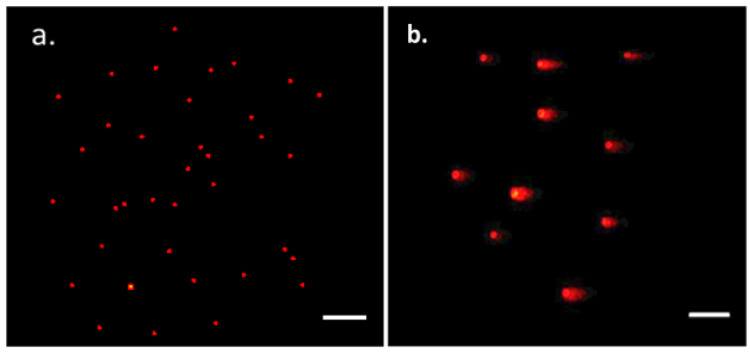
DNA fragmentation studies of PTAgNPs. (**a**) Control; (**b**) treated.

**Figure 12 plants-12-01261-f012:**
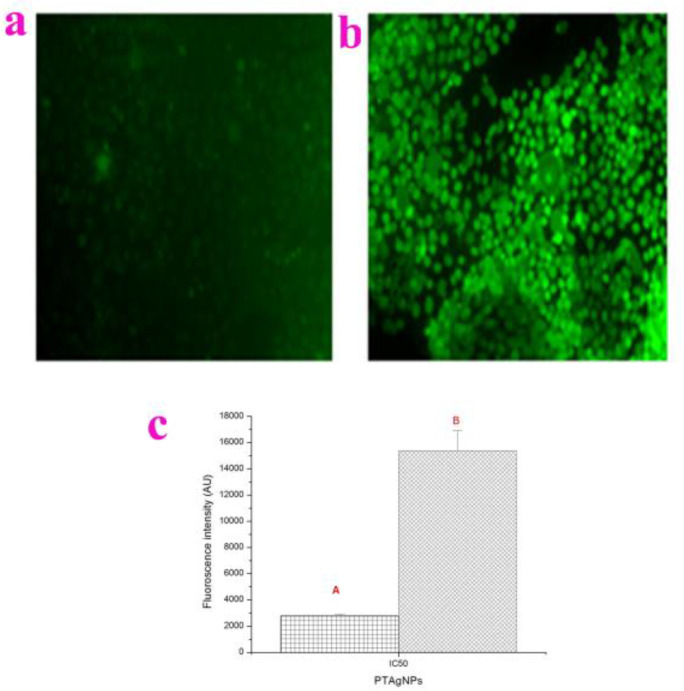
Fluorescent microscopic image of intracellular ROS analysis by DCFH-DA staining. (**a**) Control; (**b**) treated (**c**) Spectroscopic fluorescence intensity measurement of PTAgNPs treated A431 cells compared with control. The graphical representation indicates the production of ROS examined by a spectrofluorometer. A—Control cells, B—treated cells. Values are the mean SD from three independent replicate experiments (significantly different from the control group *p* < 0.05).

## Data Availability

Not applicable.

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
