# Peer review of "Polianthes tuberosa-Mediated Silver Nanoparticles from Flower Extract and Assessment of Their Antibacterial and Anticancer Potential: An In Vitro Approach"

_plants, 2023, doi:10.3390/plants12061261_

Round 1

Reviewer 1 Report

In this study, the authors synthesized and characterized stable silver nanoparticles (PTAgNPs) using Polianthes tuberosa flowers extracts. Through a series experiments, the antibacterial and anticancer activities of silver nanoparticles were proved, and its mechanism was preliminarily explored. This study was meaningful and innovative. However, some details and experiments need to be supplemented. It was recommended to publish after appropriate modification. Some specific suggestions were as follows:

1.     Line 33-34, the authors said “while having minimal toxicity towards healthy cells.”, but there was no experimental verification.

2.     Line 80-81, The authors need to elaborate on the recent progress in PTAgNPs against cancer.

3.     Line 151 and 175, The pictures and fonts were unclear and heavily distorted (Figure 2 and Figure 4).

4.     Line 196-206, Some experiments were needed to confirm the hypothesis of antibacterial mechanism of silver nanoparticles.

Author Response

In this study, the authors synthesized and characterized stable silver nanoparticles (PTAgNPs) using Polianthes tuberosa flowers extracts. Through a series experiments, the antibacterial and anticancer activities of silver nanoparticles were proved, and its mechanism was preliminarily explored. This study was meaningful and innovative. However, some details and experiments need to be supplemented. It was recommended to publish after appropriate modification. Some specific suggestions were as follows:

  1. Line 33-34,the authors said “while having minimal toxicity towards healthy cells.”, but there was no experimental verification.

Authors response: We thank the reviewer for the critical reviewing the manuscript. As the reviewer suggests the experimental verification; we remove the sentence from the line: 33-34 and added in the cytotoxicity assay results and justified in the results and discussion part.

  1. Line 80-81, The authors need to elaborate on the recent progress in PTAgNPs against cancer.

Authors’ response: We have elaborated the recent progress of AgNPs against cancer

  1. Line 151 and 175, The pictures and fonts were unclear and heavily distorted (Figure 2 and Figure 4).

Authors’ response: We humbly accept the reviewer query. For the betterment of the manuscript we have separated the fig 2 and 4 and corrected the font size

  1. Line 196-206, Some experiments were needed to confirm the hypothesis of antibacterial mechanism of silver nanoparticles.

Authors’ response: The experimental evidences with appropriate references was added to the experimental evidence of antibacterial mechanism of silver nanoparticles. 

Reviewer 2 Report

The manuscript submitted by Mousa A. Alghuthaymi, Sunita Patil ,Chandrasekaran Rajkuberan, Muthukumar Krishnan , Ushani Krishnan and Kamel A Abd-Elsalam, “POLIANTHES TUBEROSA-MEDIATED SILVER NANOPARTICLES FROM FLOWER EXTRACT AND ASSESSMENT OF THEIR ANTIBACTERIAL AND ANTICANCER POTENTIAL: AN IN VITRO APPROACH”, reports the synthesis of silver nanoparticles using Polianthes tuberosa flower as a reducing and stabilizing agent, their structural and morphological characterization, as well as their antibacterial activity and the effect on inhibiting the growth of melanoma cells.

Overall, the manuscript is systematically, clear and concise written, provides the necessary information to understand the motivation and methods described, the conclusions are consistent with the evidence and arguments presented. Here are some suggestions / remarks that should be addressed to the authors to revise and improve their work before acceptance for publishing in Plants journal:

1 – Review the references, for example, reference 20 is not the one mentioned in the text.

2- The structural characterisation of silver nanoparticles by FTIR and XRD methods are lacking in references to support the obtained results.

3- The description of figure 5 should mention that the control is gentamicin.

4- For text clarity, line 194 should mention the investigated gram positive bacteria.

5- As presented, all the experiments on silver nanoparticles regarding the anticancer activity were carried out on the cell line A 431. I consider that the authors should mention in the manuscript that the used cell line is a skin cancer cell line. Also, since the nanoparticles are to be used in treatment of human skin cancer, I strongly recommend that the authors should present in the manuscript the biocompatibility of synthetized silver nanoparticles with a normal healthy cell line.

6- Lines 261-264 “As a consequence of PTAgNPs treatment in the A431 cell line, ROS were generated in response to PTAgNPs treatment, and the treated cells significantly exhibited higher green fluorescent intensity than the control (Fig.9), while control cells were stained with less intensity.” should be revised for a better understanding of the text.

7- Lines 289-290: “The flowers were delivered to the lab soon after being collected and surface sterilized with water.” How were the flowers sterilized with water? Did the authors intended to write washed with water?

8- The English language and style presents minor spell check which requires modifications. For example:

-        - there are a lot of spaces missing between words;

-        - plant names and “in vitro” should be in italics in all manuscript;

-        - line 117 “we fixed the reaction parameters as fellow” should be “as follows”;

-        - line 200 “stricture” should be “structure”;

-        - line 252 “cell cine A341” should be “A 431”.

In conclusion, the manuscript consists in an original work with potential scientific importance and is consistent with the scope and aims of the Plants journal, so it could be accepted after major revision made by the authors.

Author Response

The manuscript submitted by Mousa A. Alghuthaymi, Sunita Patil ,Chandrasekaran Rajkuberan, Muthukumar Krishnan , Ushani Krishnan and Kamel A Abd-Elsalam, “POLIANTHES TUBEROSA-MEDIATED SILVER NANOPARTICLES FROM FLOWER EXTRACT AND ASSESSMENT OF THEIR ANTIBACTERIAL AND ANTICANCER POTENTIAL: AN IN VITRO APPROACH”, reports the synthesis of silver nanoparticles using Polianthes tuberosa flower as a reducing and stabilizing agent, their structural and morphological characterization, as well as their antibacterial activity and the effect on inhibiting the growth of melanoma cells.

Overall, the manuscript is systematically, clear and concise written, provides the necessary information to understand the motivation and methods described, the conclusions are consistent with the evidence and arguments presented. Here are some suggestions / remarks that should be addressed to the authors to revise and improve their work before acceptance for publishing in Plants journal:

1 – Review the references, for example, reference 20 is not the one mentioned in the text.

Author’s response: The reference 20 is added in the text

2- The structural characterisation of silver nanoparticles by FTIR and XRD methods are lacking in references to support the obtained results.

Author’s response: The FTIR and XRD results were supported with references in the revised manuscript

3- The description of figure 5 should mention that the control is gentamicin.

Author’s response: The control Gentamicin was mentioned in the revised manuscript.

4- For text clarity, line 194 should mention the investigated gram positive bacteria.

Author’s response: In the revised manuscript, the investigated gram positive bacteria was added in the text.

5- As presented, all the experiments on silver nanoparticles regarding the anticancer activity were carried out on the cell line A 431. I consider that the authors should mention in the manuscript that the used cell line is a skin cancer cell line. Also, since the nanoparticles are to be used in treatment of human skin cancer, I strongly recommend that the authors should present in the manuscript the biocompatibility of synthetized silver nanoparticles with a normal healthy cell line.

Author’s response: We humbly accept the reviewer query. With appropriate justification and reference we have added the biocompatible nature of synthesized silver nanoparticles in the revised manuscript.

6- Lines 261-264 “As a consequence of PTAgNPs treatment in the A431 cell line, ROS were generated in response to PTAgNPs treatment, and the treated cells significantly exhibited higher green fluorescent intensity than the control (Fig.9), while control cells were stained with less intensity.” should be revised for a better understanding of the text.

Author’s response: In the revised manuscript we have rephrased the sentence.

7- Lines 289-290: “The flowers were delivered to the lab soon after being collected and surface sterilized with water.” How were the flowers sterilized with water? Did the authors intended to write washed with water?

Author’s response: In the revised manuscript we have changed the sentence washed with water instead of surface sterilized.

8- The English language and style presents minor spell check which requires modifications. For example:

-        - there are a lot of spaces missing between words;

Author’s response: In the revised manuscript we have corrected

        - plant names and “in vitro” should be in italics in all manuscript;

Author’s response: In the revised manuscript we have corrected

        - line 117 “we fixed the reaction parameters as fellow” should be “as follows”;

Author’s response: In the revised manuscript we have corrected

        - line 200 “stricture” should be “structure”;

Author’s response: In the revised manuscript we have corrected

        - line 252 “cell cine A341” should be “A 431”.

Author’s response: In the revised manuscript we have corrected

In conclusion, the manuscript consists in an original work with potential scientific importance and is consistent with the scope and aims of the Plants journal, so it could be accepted after major revision made by the authors

Round 2

Reviewer 1 Report

The authors improved the manuscript.

Author Response

The english style and overall  minor spell check is revised and correct in the manuscript  

Reviewer 2 Report

The manuscript submitted by Mousa A. Alghuthaymi, Sunita Patil ,Chandrasekaran Rajkuberan, Muthukumar Krishnan , Ushani Krishnan and Kamel A Abd-Elsalam, “POLIANTHES TUBEROSA-MEDIATED SILVER NANOPARTICLES FROM FLOWER EXTRACT AND ASSESSMENT OF THEIR ANTIBACTERIAL AND ANTICANCER POTENTIAL: AN IN VITRO APPROACH”, has been improved, the authors kindly responded to all comments.

            There are still a few things that need to be reviewed:

- The English language still presents minor spell check which requires modifications. For example: there are some spaces missing between words; “in vitro” in title, plant names Polianthes tuberosa (line 120, 127) Pimpinella anisum (line 133) and should be in italics; line 287 – “oidative stress” should be “oxidative stress”;

- Line 332-333 - Phrase “The report of karimzadeh et al [43] proved that 332 biogenic silver nanoparticles doesn’t exert much toxicity and biocompatible to the cells.” should be revised for a better understanding of the text, and the author's name should be capitalized;

- Line 337-338 – Phrase “But howsoever, it is warranted to study the biocompatible nature of the cells in response to the exposure of silver nanoparticles.” should be revised revised for a better understanding of the text;

- Line 428 – Phrase “The flowers were dried in a hot air oven at 37°C for 7 days after sterilization.” should be modified since the flowers were not previously washed with water, not sterilized;

- In the Conclusions chapter, line 515, the authors should mention once more that the paper is describing Polianthes tuberosa-mediated silver nanoparticles from flower extract, instead of referring to generic silver nanoparticles.

          Overall, as I mentioned before, the subject of the paper is certainly interesting, so it deserves to be published after minor revisions.

Author Response

The manuscript submitted by Mousa A. Alghuthaymi, Sunita Patil ,Chandrasekaran Rajkuberan, Muthukumar Krishnan , Ushani Krishnan and Kamel A Abd-Elsalam, “POLIANTHES TUBEROSA-MEDIATED SILVER NANOPARTICLES FROM FLOWER EXTRACT AND ASSESSMENT OF THEIR ANTIBACTERIAL AND ANTICANCER POTENTIAL: AN IN VITRO APPROACH”, has been improved, the authors kindly responded to all comments.

            There are still a few things that need to be reviewed:

- The English language still presents minor spell check which requires modifications. For example: there are some spaces missing between words; “in vitro” in title, plant names Polianthes tuberosa (line 120, 127) Pimpinella anisum (line 133) and should be in italics; line 287 – “oidative stress” should be “oxidative stress”;

Authors Response: We accept the reviewer suggestion, in the revised manuscript minor spell check mistakes are corrected.

- Line 332-333 - Phrase “The report of karimzadeh et al [43] proved that 332 biogenic silver nanoparticles doesn’t exert much toxicity and biocompatible to the cells.” should be revised for a better understanding of the text, and the author's name should be capitalized;

Authors response: The sentence is rephrased in the revised manuscript

- Line 337-338 – Phrase “But howsoever, it is warranted to study the biocompatible nature of the cells in response to the exposure of silver nanoparticles.” should be revised revised for a better understanding of the text;

Authors response: The sentence is rephrased in the revised manuscript

- Line 428 – Phrase “The flowers were dried in a hot air oven at 37°C for 7 days after sterilization.” should be modified since the flowers were not previously washed with water, not sterilized;

Authors response: The sentence is modified  in the revised manuscript

 In the Conclusions chapter, line 515, the authors should mention once more that the paper is describing Polianthes tuberosa-mediated silver nanoparticles from flower extract, instead of referring to generic silver nanoparticles.

Authors response: The sentence is modified  in the revised manuscript
